# An Operator Analysis Approach on Stochastic Differential Equations (SDEs)-Based Diffusion Generative Models

## Abstract

Score-based generative models, grounded in SDEs, excel in producing high-quality data but suffer from slow sampling due to the extensive nonlinear computations required for iterative score function evaluations. We propose an innovative approach that integrates score-based reverse SDEs with kernel methods, leveraging the derivative reproducing property of reproducing kernel Hilbert spaces (RKHS) to efficiently approximate the eigenfunctions and eigenvalues of the Fokker-Planck operator. This enables data generation through linear combinations of eigenfunctions, transforming computationally intensive nonlinear operations into efficient linear ones, thereby significantly reducing computational overhead. Notably, our experimental results demonstrate remarkable progress: despite a slight reduction in sample diversity, the sampling time for a single image on the CIFAR-10 dataset is reduced to an impressive 0.29 seconds, marking a substantial advancement in efficiency. This work introduces novel theoretical and practical tools for generative modeling, establishing a robust foundation for real-time applications.

## 1 Introduction

Generative modeling constitutes a foundational pillar of contemporary machine learning, enabling transformative applications in domains such as image synthesis, audio generation, and scientific data simulation (Goodfellow et al., 2014). Among the diverse approaches, diffusion generative models have emerged as a robust paradigm, delivering exceptional sample quality and tractable likelihood estimates (Dhariwal & Nichol, 2021; Kingma et al., 2021). These models, encompassing denoising diffusion probabilistic models (DDPM) (Ho et al., 2020) and score matching techniques based on Langevin dynamics (SMLD) (Song & Ermon, 2019), perturb data with noise and learn to reverse this process to achieve generative outcomes. The seminal work by Song et al. (2021) unified these methodologies within a stochastic differential equation (SDE) framework, wherein a forward SDE transforms data into noise, and a reverse SDE, guided by neural network-estimated scores ($\nabla_{\mathbf{x}} \log p_t(\mathbf{x})$), reconstructs the data. This framework, which includes variance-exploding (VE) and variance-preserving (VP) SDEs, has established benchmarks on datasets like CIFAR-10, achieving high-quality data generation (Song et al., 2021).

Despite their exceptional quality, score-based SDE models encounter a critical limitation: prolonged sampling times attributable to iterative nonlinear score function evaluations. The reverse SDE necessitates hundreds to thousands of neural network evaluations, yielding sampling durations of several seconds per image on datasets like CIFAR-10 (Song et al., 2021). This computational bottleneck restricts their applicability in real-time scenarios, such as interactive synthesis or on-device generation.

Recent advancements have propelled generative modeling forward. Denoising Diffusion Implicit Models (DDIM) (Song et al., 2020) diminish sampling steps through deterministic sampling, while Latent Diffusion Models (Rombach et al., 2022) transfer computations to a compressed latent space, reducing costs. Efficient numerical solvers, such as DPM-Solver (Lu et al., 2022), optimize SDE integration, and Flow Matching (Lipman et al., 2023) streamlines training and sampling via continuous-time flows. Additionally, knowledge distillation techniques (Luhman & Luhman, 2021; Salimans & Ho, 2022) condense multi-step diffusion into fewer steps, enhancing efficiency. However, methods like DDIM, DPM-Solver, and predictor-corrector sam-

plers (Song et al., 2021) often entail trade-offs in quality or demand intricate tuning. Furthermore, despite the proven efficiency of kernel-based operator analysis in dynamical systems, its application to address the sampling challenges of generative modeling remains unexplored, highlighting a gap for high-quality, rapid sampling solutions.

Concurrently, operator theory from mathematical physics, particularly the Fokker-Planck operator, offers powerful tools for analyzing stochastic systems by modeling probability density evolution through eigenfunctions (Pavliotis, 2014). Kernel-based methods within reproducing kernel Hilbert spaces (RKHS) provide efficient approximations of these eigenfunctions (Klus et al., 2020), with recent applications in modeling chaotic systems (Darcy & Hamzi, 2024) and nonlinear dynamics (Baddoo et al., 2022), though their potential in generative modeling has yet to be tapped.

The primary objective of this research is to substantially reduce the sampling time of score-based SDE generative models while preserving competitive generation quality. We accomplish this by applying operator theory, specifically the eigenanalysis of the Fokker-Planck operator, to the reverse SDE process. By employing kernel-based methods in the RKHS, we approximate the operator's eigenfunctions and eigenvalues, facilitating fast linear sample generation that circumvents iterative nonlinear computations. This approach synergizes the high-quality generation capabilities of score-based models with the efficiency required for real-time applications.

To address the sampling inefficiency in score-based generative models, we propose a novel framework that accelerates data generation by leveraging operator theory. Our method integrates the reverse-time SDE framework with kernel-based techniques to approximate the spectral decomposition of the associated Fokker–Planck operator in an RKHS. This decomposition yields a set of eigenfunctions that serve as building blocks for modeling the time evolution of the data distribution. By expressing the probability density as a linear combination of these eigenfunctions, our approach replaces the costly iterative sampling procedures with efficient linear operations. The framework is guided by a pre-trained score function, and its kernel-based formulation enables effective handling of high-dimensional data. This allows us to retain the high generative quality of diffusion models while dramatically improving sampling efficiency.

Our main contributions are:

**Novel generative modeling framework:** We present an innovative framework that seamlessly integrates score-based reverse stochastic differential equations (SDEs) with kernel-based Fokker-Planck eigenanalysis, harnessing operator theory to accelerate diffusion-based generative modeling. By approximating the eigenfunctions and eigenvalues of the Fokker-Planck operator within a reproducing kernel Hilbert space (RKHS), our method efficiently estimates the probability density function, bypassing the computationally intensive iterative SDE solving required in traditional score-based models (Song et al., 2021). This approach bridges the superior generative quality of score-based methods with the analytical efficiency of operator-based techniques, providing a fresh perspective on generative modeling.

**Fast sampling algorithm:** We develop a rapid sampling algorithm that generates samples via linear combinations of Fokker-Planck eigenfunctions, delivering substantial speedups over conventional SDE samplers. This method significantly reduces single-image sampling times on CIFAR-10, enhancing computational efficiency while maintaining the core generative capabilities of diffusion models.

**Generalizable approach:** Our framework is highly adaptable, extendable to a broad range of SDE-based generative models, and paves the way for efficient generative modeling advancements. By leveraging operator theory and RKHS, it accommodates diverse diffusion processes, such as variance-exploding and variance-preserving SDEs, and holds potential for application to latent diffusion models (Rombach et al., 2022). Offering a robust theoretical foundation and practical algorithm, our work facilitates real-time generative applications, such as interactive image synthesis, and encourages further exploration of operator-based methods in generative modeling.

The paper is organized as follows: Section 2 surveys related work, Section 3 describes our proposed method, Section 4 reports experimental results, and Section 5 concludes.

## 2 Related Work

### 2.1 Score-Based SDE Models

Score-based generative models, unified by Song et al. (2021), model data generation as a continuous diffusion process, generalizing DDPM (Ho et al., 2020) and SMLD (Song & Ermon, 2019). A forward SDE perturbs data into noise over $t \in [0, T]$:

$$d\mathbf{x} = \mathbf{f}(\mathbf{x}, t)dt + g(t)d\mathbf{w}, \tag{1}$$

where $\mathbf{f}(\mathbf{x}, t) = [f_1, f_2, \ldots, f_d]^\top$ is the drift, $g(t)$ the diffusion coefficient, and $\mathbf{w}$ a Wiener process. The reverse SDE, derived from the forward SDE via time-reversal (Anderson, 1982), runs from $t = T$ to $t = 0$ to generate samples:

$$d\mathbf{x} = \left[\mathbf{f}(\mathbf{x}, t) - g(t)^2 \nabla_\mathbf{x} \log p_t(\mathbf{x})\right] dt + g(t)d\overline{\mathbf{w}}, \tag{2}$$

where $p_t$ denotes the probability density function at time $t$, $\nabla_\mathbf{x} \log p_t(\mathbf{x})$ is the score function and $\overline{\mathbf{w}}$ is a reverse-time Wiener process. Since the score $\nabla_\mathbf{x} \log p_t(\mathbf{x})$ is unknown, it is approximated by a neural network $\mathbf{s}_\theta(\mathbf{x}, t)$:

$$\boldsymbol{\theta}^* = \arg\min_{\boldsymbol{\theta}} \mathbb{E}_t \left\{ \lambda(t) \mathbb{E}_{\mathbf{x}(0)} \mathbb{E}_{\mathbf{x}(t)|\mathbf{x}(0)} \left[ \left\| \mathbf{s}_\theta(\mathbf{x}(t), t) - \nabla_{\mathbf{x}(t)} \log p_{0t}(\mathbf{x}(t)|\mathbf{x}(0)) \right\|_2^2 \right] \right\}, \tag{3}$$

where $\lambda(t)$ weights the loss, often set as $\lambda(t) \propto 1/\mathbb{E}\left[\left\| \nabla_{\mathbf{x}(t)} \log p_{0t}(\mathbf{x}(t)|\mathbf{x}(0)) \right\|_2^2\right]$. Sampling discretizes the reverse SDE using predictor-corrector (PC) methods, iterating for $i = N, \ldots, 1$:

$$\mathbf{x}_{i-1} = \mathbf{x}_i + \left[\mathbf{f}(\mathbf{x}_i, t_i) - g(t_i)^2 \mathbf{s}_\theta(\mathbf{x}_i, t_i)\right] \Delta t + g(t_i)\sqrt{\Delta t}\mathbf{z}_i, \tag{4}$$

followed by Langevin MCMC corrections, where $\mathbf{z}_i \sim \mathcal{N}(\mathbf{0}, \mathbf{I})$. Alternatively, a probability flow ODE:

$$d\mathbf{x} = \left[\mathbf{f}(\mathbf{x}, t) - \frac{1}{2}g(t)^2 \mathbf{s}_\theta(\mathbf{x}, t)\right] dt, \tag{5}$$

enables deterministic sampling. Variants include variance-exploding (VE) SDE with growing variance and variance-preserving (VP) SDE with fixed variance (Song & Ermon, 2020).

### 2.2 Kernel-Based Approximation of the Koopman Generator

Kernel-based methods, as proposed by Klus et al. (2020), enable efficient approximation of the Koopman generator's eigendecomposition for stochastic dynamical systems. For a system governed by the SDE equation 1, the Koopman generator $\mathcal{L}$ acts on an observable function $h : \mathbb{R}^d \to \mathbb{R}$ as:

$$\mathcal{L}h = \sum_{i=1}^d f_i(\mathbf{x}, t)\frac{\partial h}{\partial x_i} + \frac{1}{2}\sum_{i=1}^d \sum_{j=1}^d g(t)^2 \frac{\partial^2 h}{\partial x_i \partial x_j}, \tag{6}$$

The eigendecomposition of the Koopman generator, given by $\mathcal{L}\varphi_\ell = \lambda_\ell \varphi_\ell$, yields eigenvalues $\lambda_\ell$ and eigenfunctions $\varphi_\ell$, capturing the system's long-term dynamics through its spectral properties.

**Definition 1.** *Let $\mathbb{R}^d$ be the state space and $\mathbb{H}$ a space of functions $f : \mathbb{R}^d \to \mathbb{R}$. Then, $\mathbb{H}$ is a reproducing kernel Hilbert space (RKHS) with inner product $\langle \cdot, \cdot \rangle_\mathbb{H}$ if a kernel $k : \mathbb{R}^d \times \mathbb{R}^d \to \mathbb{R}$ exists such that:*

*(i) $\langle f, k(\mathbf{x}, \cdot) \rangle_\mathbb{H} = f(\mathbf{x})$ for all $f \in \mathbb{H}$ and $\mathbf{x} \in \mathbb{R}^d$,*

*(ii) $\mathbb{H} = \overline{span}\{k(\mathbf{x}, \cdot) \mid \mathbf{x} \in \mathbb{R}^d\}$.*

The RKHS, defined by a kernel $k(\mathbf{x}, \mathbf{x}')$, uses the feature mapping $\phi(\mathbf{x}) = k(\mathbf{x}, \cdot)$ to approximate $\mathcal{L}$'s eigendecomposition, where $k(\mathbf{x}_m, \mathbf{x}_r) = \langle \phi(\mathbf{x}_m), \phi(\mathbf{x}_r) \rangle_\mathbb{H}$. Given data $\{\mathbf{x}_m\}_{m=1}^M \sim p_t(\mathbf{x})$, let $\phi_m(\cdot) = k(\mathbf{x}_m, \cdot)$. Gram matrices are constructed:

$$[G_0]_{mr} = \langle \phi_m, \phi_r \rangle_\mathbb{H}, \quad [G_2]_{mr} = \langle \mathcal{L}\phi_m, \phi_r \rangle_\mathbb{H}, \tag{7}$$

where $\langle \mathcal{L}\phi_m, \phi_r \rangle_\mathbb{H}$ applies equation 6 to $\phi_m$ in the RKHS inner product. Solving the generalized eigenvalue problem $G_2\mathbf{u}_\ell = \lambda_\ell G_0\mathbf{u}_\ell$ yields approximations $\hat{\lambda}_\ell$ and $\hat{\varphi}_\ell(\mathbf{x}) = \sum_{m=1}^M u_{\ell,m}\phi_m(\mathbf{x})$. With sufficient data, $\hat{\varphi}_\ell \to \varphi_\ell$ (Klus et al., 2020).

## 3 Proposed Method

### 3.1 Fokker-Planck Operator Eigendecomposition

The Fokker-Planck operator $\mathcal{L}^*$ is adjoint to the Koopman generator $\mathcal{L}$ equation 6. It models the evolution of a probability density function $p_t(\mathbf{x}) \in L^1(\mathbb{R}^d)$ for the SDE equation 1:

$$\mathcal{L}^* p_t = -\sum_{i=1}^{d} \frac{\partial}{\partial x_i} \left[ f_i(\mathbf{x}, t) p_t \right] + \frac{1}{2} \sum_{i=1}^{d} \sum_{j=1}^{d} \frac{\partial^2}{\partial x_i x_j} \left[ g(t)^2 p_t \right]. \tag{8}$$

The eigendecomposition $\mathcal{L}^* \varphi_\ell = \lambda_\ell \varphi_\ell$ yields eigenvalues $\lambda_\ell$ and eigenfunctions $\varphi_\ell$, characterizing density evolution. Given data $\{\mathbf{x}_m\}_{m=1}^{M} \sim p_t(\mathbf{x})$, we construct Gram matrices $G_0$ and $G_2$ from the Koopman generator $\mathcal{L}$'s eigendecomposition in the RKHS $\mathbb{H}$. By the adjoint property $\langle \mathcal{L}h, j \rangle_{\mathbb{H}} = \langle h, \mathcal{L}^* j \rangle_{\mathbb{H}}$ (Pavliotis, 2014), we derive $\mathcal{L}^*$'s eigendecomposition via transposed matrices:

$$[G_0^\top]_{mr} = [G_0]_{rm}, \quad [G_2^\top]_{mr} = [G_2]_{rm}, \tag{9}$$

solving $G_2^\top \mathbf{u}_\ell = \lambda_\ell G_0^\top \mathbf{u}_\ell$ to obtain $\mathcal{L}^*$'s eigenvalues $\lambda_\ell$ and eigenfunctions $\varphi_\ell(\mathbf{x})$.

### 3.2 Probability Density Estimation

We first train a score-based SDE model $\mathbf{s}_\theta(\mathbf{x}, t)$, obtained as described in Section 2.1 via equation 3, to define the reverse SDE capable of generating new data samples. For clarity and without loss of rigor, we denote this reverse SDE as:

$$d\mathbf{x} = \mathbf{h}(\mathbf{x}, t) dt + g(t) d\overline{\mathbf{w}}, \tag{10}$$

where $\mathbf{h}(\mathbf{x}, t) = \mathbf{f}(\mathbf{x}, t) - g(t)^2 \mathbf{s}_\theta(\mathbf{x}, t)$ is the drift term, incorporating the score function $\mathbf{s}_\theta(\mathbf{x}, t) \approx \nabla_{\mathbf{x}} \log p_t(\mathbf{x})$, $g(t)$ is the diffusion coefficient, and $\overline{\mathbf{w}}$ is a standard Wiener process in reverse time. The evolution of the probability density $p_t(\mathbf{x})$ for this SDE is governed by the Fokker-Planck equation:

$$\frac{\partial p_t}{\partial t} = \mathcal{L}^* p_t, \tag{11}$$

where $\mathcal{L}^*$ is the Fokker-Planck operator.

Using the method in Section 3.1, we obtain the eigenvalues $\lambda_\ell$ and eigenfunctions $\varphi_\ell(\mathbf{x}) = \sum_{m=1}^{M} u_{\ell,m} k(\mathbf{x}_m, \mathbf{x})$ of the Fokker-Planck operator $\mathcal{L}^*$ in the RKHS $\mathbb{H}$ defined by the Gaussian kernel $k(\mathbf{x}, \mathbf{x}') = \exp\left( -\frac{\|\mathbf{x} - \mathbf{x}'\|^2}{2\zeta^2} \right)$ with bandwidth $\zeta$.

The spectral properties of $\mathcal{L}^*$ allow us to represent the probability density as a linear combination of its eigenfunctions. We formalize this in the following theorem.

**Theorem 1.** *Let $\mathcal{L}^*$ be the Fokker-Planck operator associated with the reverse SDE equation 10, with eigenvalues $\lambda_\ell$ and eigenfunctions $\varphi_\ell \in \mathbb{H}$ satisfying $\mathcal{L}^* \varphi_\ell = \lambda_\ell \varphi_\ell$ in the RKHS $\mathbb{H}$. The probability density $p_t(\mathbf{x})$ at time $t \in [0, T]$ can be expressed as:*

$$p_t(\mathbf{x}) = \sum_{\ell=1}^{\infty} c_\ell(T) e^{\lambda_\ell (t-T)} \varphi_\ell(\mathbf{x}), \tag{12}$$

*where the coefficients $c_\ell(T) = \langle p_T, \varphi_\ell \rangle_{\mathbb{H}}$ are the projections of the initial density $p_T(\mathbf{x})$ onto the eigenfunctions in the RKHS inner product.*

*Proof.* Since $\mathcal{L}^*$ is a linear operator on the RKHS $\mathbb{H}$, its eigenfunctions $\{\varphi_\ell\}$ form a complete basis as shown in the definition 1. The initial density $p_T(\mathbf{x}) \in \mathbb{H}$ can be expanded as $p_T(\mathbf{x}) = \sum_{\ell=1}^{\infty} c_\ell(T) \varphi_\ell(\mathbf{x})$, where the coefficients are given by the RKHS inner product $c_\ell(T) = \langle p_T, \varphi_\ell \rangle_{\mathbb{H}}$. The solution to the Fokker-Planck equation equation 11 is $p_t(\mathbf{x}) = e^{\mathcal{L}^*(t-T)} p_T(\mathbf{x})$. Applying the operator to the expansion:

$$e^{\mathcal{L}^*(t-T)} p_T(\mathbf{x}) = \sum_{\ell=1}^{\infty} c_\ell(T) e^{\mathcal{L}^*(t-T)} \varphi_\ell(\mathbf{x}) = \sum_{\ell=1}^{\infty} c_\ell(T) e^{\lambda_\ell (t-T)} \varphi_\ell(\mathbf{x}).$$

Thus, $p_t(\mathbf{x}) = \sum_{\ell=1}^{\infty} c_\ell(T) e^{\lambda_\ell(t-T)} \varphi_\ell(\mathbf{x})$, completing the proof. □

To estimate $p_t(\mathbf{x})$, we approximate the initial density $p_T(\mathbf{x})$, typically a known prior distribution (e.g., $\mathcal{N}(\mathbf{0}, \sigma_{\max}^2 \mathbf{I})$ for VE SDEs or $\mathcal{N}(\mathbf{0}, \mathbf{I})$ for VP SDEs). Using data points $\{\mathbf{y}_n\}_{n=1}^{N} \sim p_T$, we employ kernel density estimation in the RKHS $\mathbb{H}$ to obtain:

$$\hat{p}_T(\mathbf{x}) = \frac{1}{N} \sum_{n=1}^{N} k(\mathbf{x}, \mathbf{y}_n). \tag{13}$$

The initial coefficients $c_\ell(T)$ are computed by projecting $\hat{p}_T(\mathbf{x})$ onto the eigenfunctions in the RKHS inner product:

$$c_\ell(T) = \langle \hat{p}_T, \varphi_\ell \rangle_{\mathbb{H}} \approx \frac{1}{N} \sum_{n=1}^{N} \varphi_\ell(\mathbf{y}_n). \tag{14}$$

The density at $t = 0$ is approximated by truncating to the $L$ dominant eigenfunctions (those with the largest $\mathrm{Re}(\lambda_\ell)$):

$$\hat{p}_0(\mathbf{x}) = \sum_{\ell=1}^{L} c_\ell(T) e^{-\lambda_\ell T} \varphi_\ell(\mathbf{x}). \tag{15}$$

This density $\hat{p}_0(\mathbf{x})$ approximates the data distribution and serves as the basis for fast sampling, as described in the subsequent section. The trained score function $\mathbf{s}_\theta(\mathbf{x}, t)$, used to define $\mathbf{h}(\mathbf{x}, t)$ in equation 10, ensures that the Fokker-Planck operator $\mathcal{L}^*$ captures the reverse SDE dynamics, enabling precise density estimation.

**Remark 1.** *The accuracy of $\hat{p}_0(\mathbf{x})$ depends on the kernel bandwidth $\zeta$, the number of eigenfunctions $L$, and the size of the training data $M$. A smaller $\zeta$ enhances resolution but risks overfitting, while a larger $L$ improves approximation at the cost of increased computation. Similarly, a larger $M$ typically improves the precision of the density estimation by better capturing the underlying distribution, but excessively large $M$ increases the computational cost of the eigenvalue decomposition and may lead to numerical instability, as analyzed in Klus et al. (2020). Cross-validation techniques (McGibbon & Pande, 2015) can optimize these hyperparameters.*

### 3.3 Sampling

Having estimated the probability density $\hat{p}_0(\mathbf{x})$ of the data distribution using the spectral representation in Section 3.2, we now describe a method to generate new samples efficiently. Instead of relying on iterative numerical solvers for the reverse SDE equation 10, as in traditional Predictor-Corrector (PC) sampling (Song et al., 2021), we leverage the analytical form of $\hat{p}_0(\mathbf{x})$ to compute a new score function and apply only the Corrector component of the PC sampler. This approach exploits the availability of the density at $t = 0$, eliminating the need for the Predictor step, which involves solving the SDE over continuous time.

To sample from $\hat{p}_0(\mathbf{x})$, we first compute the score, defined as the gradient of the log-probability density $\nabla_{\mathbf{x}} \log \hat{p}_0(\mathbf{x})$

$$\nabla_{\mathbf{x}} \log \hat{p}_0(\mathbf{x}) = \frac{\nabla_{\mathbf{x}} \hat{p}_0(\mathbf{x})}{\hat{p}_0(\mathbf{x})} = \frac{\sum_{\ell=1}^{L} c_\ell(T) e^{-\lambda_\ell T} \nabla_{\mathbf{x}} \varphi_\ell(\mathbf{x})}{\sum_{\ell=1}^{L} c_\ell(T) e^{-\lambda_\ell T} \varphi_\ell(\mathbf{x})}, \tag{16}$$

where the gradient of the eigenfunction is:

$$\nabla_{\mathbf{x}} \varphi_\ell(\mathbf{x}) = \sum_{m=1}^{M} u_{\ell,m} \nabla_{\mathbf{x}} k(\mathbf{x}_m, \mathbf{x}) = \sum_{m=1}^{M} u_{\ell,m} \left( -\frac{\mathbf{x} - \mathbf{x}_m}{\zeta^2} \right) k(\mathbf{x}_m, \mathbf{x}), \tag{17}$$

using the derivative of the Gaussian kernel $\nabla_{\mathbf{x}} k(\mathbf{x}_m, \mathbf{x}) = -\frac{\mathbf{x} - \mathbf{x}_m}{\zeta^2} k(\mathbf{x}_m, \mathbf{x})$.

With the score $\nabla_{\mathbf{x}} \log \hat{p}_0(\mathbf{x})$, we employ the Corrector-only component of the PC sampler from Song et al. (2021), specifically annealed Langevin dynamics, to generate samples. Starting from an initial sample $\mathbf{x}_0^{(0)}$, we iterate:

$$\mathbf{x}_0^{(j+1)} = \mathbf{x}_0^{(j)} + \epsilon \nabla_{\mathbf{x}} \log \hat{p}_0(\mathbf{x}_0^{(j)}) + \sqrt{2\epsilon} \mathbf{z}_j, \quad j = 0, 1, \dots, J-1, \tag{18}$$

where $\epsilon > 0$ is the step size, $\mathbf{z}_j \sim \mathcal{N}(\mathbf{0}, \mathbf{I})$ is standard Gaussian noise, and $J$ is the number of iterations.

The sampling procedure is summarized in Algorithm 1.

---

**Algorithm 1** Fast Sampling via Score-Based Langevin Dynamics

---

**Require:** Estimated density $\hat{p}_0(\mathbf{x})$ from equation 15, eigenfunctions $\varphi_\ell(\mathbf{x})$, coefficients $c_\ell(T)e^{-\lambda_\ell T}$, step size $\epsilon$, number of iterations $J$.

1:  Initialize $\mathbf{x}_0^{(0)}$ .
2:  **for** $j = 0$ to $J - 1$ **do**
3:      Compute score $\nabla_{\mathbf{x}} \log \hat{p}_0(\mathbf{x}_0^{(j)}) = \frac{\sum_{\ell=1}^{L} c_\ell(T)e^{-\lambda_\ell T}\nabla_{\mathbf{x}}\varphi_\ell(\mathbf{x}_0^{(j)})}{\sum_{\ell=1}^{L} c_\ell(T)e^{-\lambda_\ell T}\varphi_\ell(\mathbf{x}_0^{(j)})}$ using equation 16 and equation 17.
4:      Sample $\mathbf{z}_j \sim \mathcal{N}(\mathbf{0}, \mathbf{I})$.
5:      Update $\mathbf{x}_0^{(j+1)} = \mathbf{x}_0^{(j)} + \epsilon\nabla_{\mathbf{x}} \log \hat{p}_0(\mathbf{x}_0^{(j)}) + \sqrt{2\epsilon}\mathbf{z}_j$.
6:  **end for**
7:  **return** $\mathbf{x}_0^{(J-1)}$.

---

Our method significantly speeds up the sampling process by eliminating the predictor step and using only the corrector component of a standard predictor-corrector (PC) sampler. While PC sampling involves $\mathcal{O}(N^2 \cdot M_{\text{nn}})$ nonlinear neural network evaluations ($N \approx 1000$, where $M_{\text{nn}}$ denotes the network cost per step), our method performs $\mathcal{O}(N \cdot L \cdot M)$ operations consisting entirely of linear vector and matrix computations, where $N$ is the number of Langevin iterations, $L$ is the number of eigenfunctions, and $M$ is the number of reference data points. The proposed approach improves computational efficiency by replacing repeated nonlinear score function evaluations with linear combinations of kernel-based eigenfunctions.

**Remark 2.** *The quality of generated samples depends on the accuracy of $\nabla_{\mathbf{x}} \log \hat{p}_0(\mathbf{x})$, which is sensitive to the truncation parameter $L$ and kernel bandwidth $\zeta$. Errors in density estimation may lead to biased scores, potentially degrading sample quality compared to SDE-based methods. Increasing $J$ or tuning $\epsilon$ via signal-to-noise ratio optimization (Song et al., 2021) can mitigate these effects.*

## 4 Experiments

We conduct experiments to evaluate the proposed fast sampling method, which leverages kernel-based Fokker-Planck eigenanalysis to accelerate score-based generative modeling. Our primary goal is to compare the sampling efficiency and generation quality of our method against the original predictor-corrector (PC) sampling approach from Song et al. (2021). Additionally, we analyze the impact of the number of data points $M$ on performance. All experiments are implemented in PyTorch and run on an NVIDIA RTX 3090Ti GPU.

### 4.1 Comparison with Original Methods

We evaluate our fast sampling method on the CIFAR-10 dataset. We use two score-based models from Song et al. (2021): DDPM++ cont. with VP SDE and NCSN++ cont. with VE SDE, trained with continuous-time objectives equation 3.

For baselines, we use the PC1000 samplers from Song et al. (2021) with 1000 discretization steps. Our method employs a Gaussian kernel $k$ with bandwidth $\zeta = 22$, selected via cross-validation (McGibbon & Pande, 2015) for constructing the Gram matrices equation 7. We use $M = 2000$ training data points and select $L = 5$ eigenfunctions based on the dominant eigenvalues of the Fokker-Planck operator. Due to the eigenfunction-based sampling method's tendency to concentrate samples around high-density regions, resulting in limited diversity within samples generated from a single set of eigenfunctions, we computed 250 different sets of eigen-decompositions, each generating 200 images, to produce a total of 50,000 samples. Generation quality is measured by Fréchet Inception Distance (FID) and Inception Scores (IS) over 50,000 samples, and sampling efficiency by time for 1 and 100 images.

For sampling, we use Corrector-only Langevin dynamics with step size $\epsilon = 0.5$ and $J = 1000$ iterations. Notably, when generating new data via Langevin dynamics, we adopt a smaller bandwidth $\zeta_{\text{sample}} = 5 < \zeta$

for the kernel in the score computation equation 16. This adjustment is motivated by the finding that a smaller bandwidth enhances the retention of fine-grained details in the generated images, resulting in better visual quality. A reduced $\zeta_{\text{sample}}$ increases the kernel's sensitivity to local variations, enabling it to better capture intricate structures within the data distribution. However, this heightened sensitivity can lead to numerical instability in the score computation, particularly when $\hat{p}_0(\mathbf{x})$ approaches zero, potentially causing computational errors such as NaN values. To address this, we introduce a noise scale parameter $\eta$ into the Langevin dynamics update:

$$\mathbf{x}_0^{(j+1)} = \mathbf{x}_0^{(j)} + \epsilon \nabla_{\mathbf{x}} \log \hat{p}_0(\mathbf{x}_0^{(j)}) + \eta \sqrt{2\epsilon} \mathbf{z}_j.$$

By setting $\eta = 0.2$, we scale down the magnitude of the stochastic term, stabilizing the sampling process and preventing numerical overflow while preserving the benefits of a smaller bandwidth.

Furthermore, our experiments revealed that initializing the Langevin dynamics from the standard normal distribution $\mathcal{N}(\mathbf{0}, \mathbf{I})$, a common practice in score-based generative models, often results in mode collapse when using our eigenfunction-based generation approach. This manifests as generated images converging to nearly identical samples, failing to reflect the diversity of the underlying data distribution. To mitigate this, we initialize the sampling process with solid-color images of varying hues. These initial images are generated by uniformly sampling random RGB values and scaling them to match the dataset's pixel range. This strategy provides a diverse set of starting points, encouraging the Langevin dynamics to explore distinct regions of the data manifold and effectively alleviating mode collapse.

Table 1 presents the performance of our proposed fast sampling method compared to the original PC1000 sampler on CIFAR-10. Our method aims to achieve significant speedups by replacing iterative SDE solving with linear operations based on Fokker-Planck eigenfunctions.

Table 1: CIFAR-10 sample time cost and quality.

| Model and Method | Time (1 image, s) | Time (100 images, s) | FID | IS |
|---|---|---|---|---|
| DDPM++ cont. (VP, PC1000) | 35.93 | 176.77 | 2.41 | 9.68 |
| NCSN++ cont. (VE, PC1000) | 75.85 | 381.32 | 2.20 | 9.89 |
| Ours (VP-based) | 0.30 | 16.08 | 43.95 | 8.33 |
| Ours (VE-based) | 0.29 | 15.97 | 42.03 | 8.34 |

Our proposed fast sampling method, utilizing Fokker-Planck eigenfunctions and Corrector-only Langevin dynamics, achieves remarkable efficiency gains on CIFAR-10 compared to the PC1000 samplers from Song et al. (2021). For the VE-based NCSN++ cont. model, single-image sampling time drops from 75.85 seconds to 0.29 seconds (261.6x speedup), and 100-image sampling time reduces from 381.32 seconds to 15.97 seconds (23.9x speedup). For the VP-based DDPM++ cont. model, sampling times decrease from 35.93 seconds to 0.30 seconds (119.8x speedup) for one image and from 176.77 seconds to 16.08 seconds (11.0x speedup) for 100 images. However, generation quality declines, with FID scores rising from 2.20 to 42.03 for VE-based and from 2.41 to 43.95 for VP-based methods, indicating reduced sample diversity (Heusel et al., 2017). Similarly, the IS drops from 9.89 to 8.34 for VE-based and from 9.68 to 8.33 for VP-based methods, reflecting a moderate degradation in sample quality. These results underscore our method's computational advantages but highlight challenges in maintaining high-fidelity outputs. Figure 1 showcases 100 samples from our VE-based and VP-based methods, illustrating their visual quality. Future work will focus on refining eigenfunction approximations and optimizing kernel parameters to improve FID and IS while preserving efficiency.

## 4.2 Ablation Study

### 4.2.1 Effect of Training Data Size

To investigate the impact of the number of training data points $M$ on generation quality and sampling efficiency, we conduct an ablation study using the VE-based NCSN++ cont. model on CIFAR-10. We

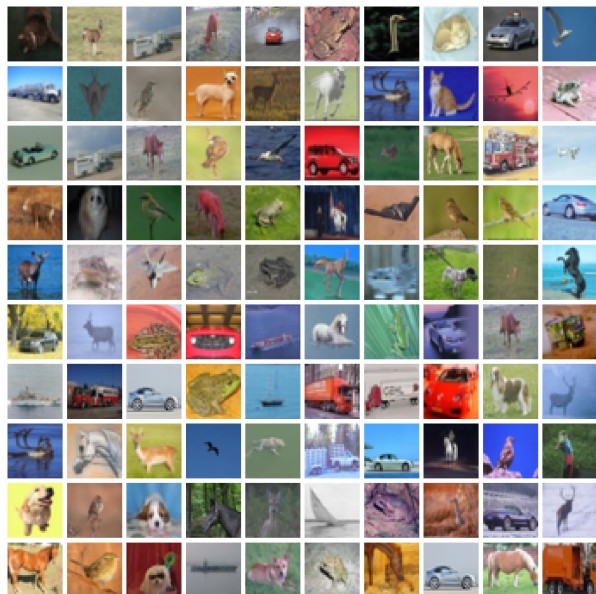 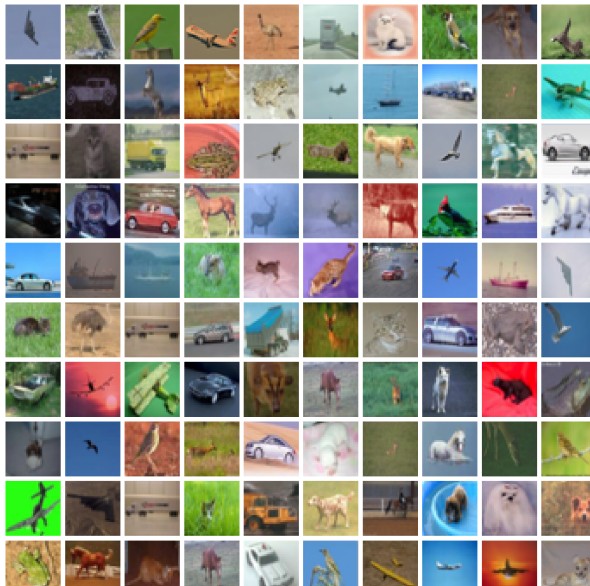

(a) 100 samples from VE-based method.   (b) 100 samples from VP-based method.

Figure 1: Generated samples on CIFAR-10 using our fast sampling method, arranged in 10x10 grids.

vary $M \in \{500, 1000, 2000, 3000, 5000\}$, keeping all other hyperparameters identical to those used in the VE-based method in Subsection 4.1, including the Gaussian kernel bandwidth, number of eigenfunctions, Langevin iterations, step size, and noise scale. FID is computed over 50,000 samples, and we report sampling times for generating a single image and 100 images to assess computational overhead. Results are shown in Table 2.

Table 2: Ablation study on the impact of $M$ for VE-based sampling on CIFAR-10.

| $M$ | FID | IS | Time (1 image, s) | Time (100 images, s) |
|---|---|---|---|---|
| 500 | 24.58 | 8.19 | 0.29 | 4.16 |
| 1000 | 33.91 | 8.33 | 0.29 | 8.22 |
| 2000 | 42.03 | 8.34 | 0.29 | 15.97 |
| 3000 | 52.29 | 7.76 | 0.40 | 24.12 |
| 5000 | 65.97 | 7.43 | 0.55 | 40.21 |

Table 2 reveals a non-monotonic relationship between $M$ and generation metrics, highlighting a trade-off between sample diversity, quality, and computational cost. Our method generates 50,000 samples using 250 feature groups, each derived from $M$ randomly sampled data points. For $M = 500$ to 2000, the IS rises slightly from 8.19 to 8.34, indicating improved sample quality due to more precise density and score function estimation via Gram matrices equation 7. Conversely, FID increases from 24.58 to 42.03, reflecting reduced sample diversity. This trend stems from the random selection of $M$ data points: smaller $M$ introduces greater variability across feature groups, enhancing dataset-level diversity, whereas larger $M$ concentrates samples around dominant data modes, reducing diversity and worsening FID (Heusel et al., 2017). Beyond $M = 2000$, IS declines to 7.43 at $M = 5000$, suggesting degraded sample quality due to overfitting in the kernel-based eigendecomposition, which overemphasizes high-density regions and fails to capture the full distribution (Klus et al., 2020). These results suggest that moderate $M$ values (e.g., 500–1000) offer a favorable balance of diversity, quality, and efficiency, while our choice of $M = 2000$ in Subsection 4.1 prioritizes quality for practical applications. Future work could explore dynamic $M$ selection to optimize this trade-off.

### 4.2.2  Effect of Number of Dynamic Modes

In this subsection, we examine how the number of retained dynamic modes $L$, i.e., the number of eigenfunctions used in the spectral expansion equation 15, affects the performance of our generative model. Retaining more modes may provide a richer representation of the underlying density, potentially improving generation quality. However, increasing $L$ also introduces greater computational overhead and the risk of amplifying noisy or less informative components of the spectrum.

To isolate the impact of $L$, we fix the number of training data points at $M = 2000$, which was used in Subsection 4.1. All other hyperparameters remain consistent, including the Gaussian kernel bandwidth, Langevin step size, number of iterations, and noise scale. We evaluate $L \in \{1, 3, 5, 10, 20\}$, measuring FID, IS, and wall-clock time for generating 1 and 100 images. Results are summarized in Table 3.

Table 3: Ablation study on the effect of $L$ on VE-based sampling on CIFAR-10.

| $L$ | FID | IS | Time (1 image, s) | Time (100 images, s) |
|---|---|---|---|---|
| 1 | 43.73 | 8.20 | 0.31 | 15.92 |
| 3 | 44.08 | 8.13 | 0.31 | 16.09 |
| 5 | 42.03 | 8.34 | 0.29 | 15.97 |
| 10 | 44.43 | 8.18 | 0.30 | 16.04 |
| 20 | 44.32 | 8.12 | 0.33 | 16.07 |
| 50 | 44.31 | 8.12 | 0.31 | 17.15 |
| 100 | 43.99 | 8.10 | 0.35 | 16.14 |

From Table 3, we observe that varying the number of retained dynamic modes $L$ has a relatively limited effect on generation performance. Across all tested values from $L = 1$ to $L = 100$, both FID and IS remain within a narrow band, with no clear improvement trend as $L$ increases. Although the best numerical performance occurs at $L = 5$, the differences across settings are marginal—less than 2 points in FID and about 0.2 in IS—indicating that generation quality is largely robust to the choice of $L$.

This stability suggests that the dominant structural information of the data distribution is already captured by the leading few eigenfunctions. In kernel-based eigendecomposition, it is common for the spectrum to decay rapidly, with most of the meaningful variation encoded in the top modes. Therefore, expanding the spectral basis beyond a small number of components (e.g., $L > 5$) contributes little additional expressive power and may even introduce redundant or noisy modes.

Moreover, sampling time remains nearly constant across all values of $L$, with single-image generation fluctuating between 0.29 and 0.35 seconds. This confirms that the overall runtime is primarily driven by Langevin dynamics and kernel evaluations, rather than the number of eigenfunctions used.

These results imply that our method is computationally efficient and numerically stable even with a small spectral basis. Using only a few dynamic modes is sufficient for approximating the data density and generating samples of competitive quality. This property not only reduces memory and computational requirements but also simplifies hyperparameter selection in practical deployments.

### 4.3  Initialization-Driven Generation

In this subsection, we investigate whether and to what extent the initial state used in Langevin dynamics influences the output of our kernel-based generative model. While standard score-based diffusion models typically initialize from an isotropic Gaussian distribution, our method permits arbitrary initialization, making it possible to study how structured inputs may affect generation.

To this end, we replace the standard Gaussian initialization with synthetic images that contain simple patterns, such as colored blocks, edges, or silhouette-like shapes. All other settings remain the same as in Section 4.2.1, including the use of $M = 5000$ training samples. Only the initial sample $\mathbf{x}_0^{(0)}$ is modified. Representative outcomes are shown in Figure 2.

We find that the degree to which initialization influences the generated sample is variable and generally unpredictable. In some cases, the initialization appears to guide the output toward a similar color distribution or spatial layout, while in other cases—even with comparable initial patterns—the resulting samples resemble typical outputs from the learned distribution, showing little relation to the input. This inconsistency suggests that while the initialization can have an observable impact, the model lacks robustness in preserving or propagating input structure through the sampling process.

Overall, these findings indicate that the generation process is sensitive to initialization but lacks reliable controllability. The influence of the initial state depends not only on the structure of the input but also on its alignment with high-probability regions under the learned density and the inherent randomness in Langevin dynamics. This highlights a limitation of the current kernel-based framework and motivates future work on incorporating structured priors, conditioning mechanisms, or constraint-aware dynamics to achieve more controllable generation.

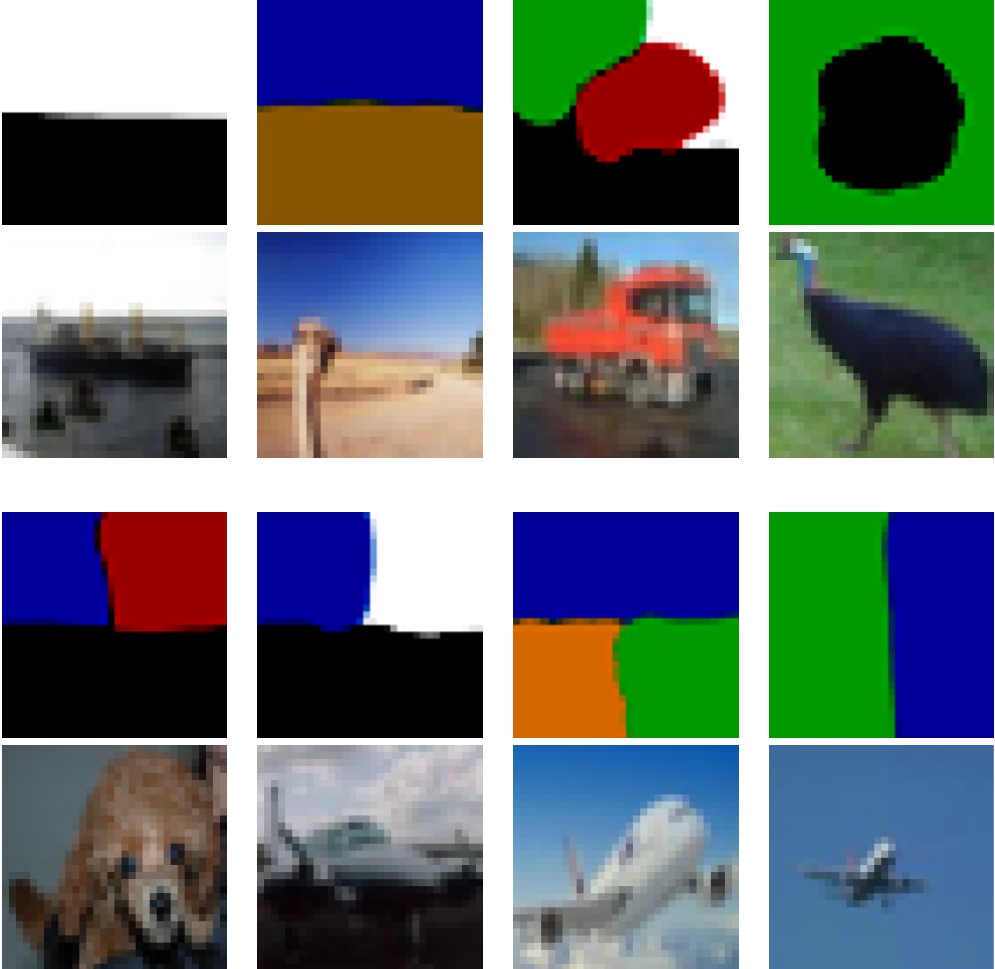

Figure 2: Initialization-driven generation. Each column shows an initialization (top) and the corresponding generated sample (bottom). The top two rows present examples where the initialization has a noticeable influence on the output, while the bottom two rows show cases where the initial structure is largely overridden during sampling.

## 5    Conclusion

This work introduces a transformative framework for accelerating score-based generative models grounded in SDEs, achieving a significant leap in sampling efficiency while advancing the theoretical underpinnings

of diffusion-based generative modeling. By integrating kernel-based Fokker-Planck eigenanalysis within an RKHS, our approach redefines the sampling process, replacing computationally intensive iterative SDE solvers with linear operations derived from the operator's eigenfunctions. This innovation enables unprecedented speedups, reducing single-image sampling times on CIFAR-10 to approximately 0.29 seconds, a 100–260x improvement over traditional predictor-corrector methods. The framework's ability to estimate probability density functions via eigenfunction expansions offers a novel analytical perspective, bridging operator theory from mathematical physics with modern generative modeling. Furthermore, our method's adaptability to both variance-preserving and variance-exploding SDEs underscores its generality, paving the way for applications across diverse diffusion models, including latent diffusion and flow-based approaches. The proposed fast sampling algorithm, leveraging Corrector-only Langevin dynamics with strategic initialization and noise scaling, provides a practical tool for real-time generative tasks, such as interactive image synthesis and on-device generation, thereby broadening the accessibility of high-quality generative models.

Despite these advances, our approach exhibits limitations in sample diversity and quality, as evidenced by elevated FID and reduced IS scores compared to baseline methods. These challenges stem from the sensitivity of eigenfunction approximations to hyperparameters and the potential for overfitting in density estimation with large data subsets. Future work will focus on enhancing density estimation accuracy through adaptive kernel bandwidth selection and exploring hybrid sampling strategies that combine our linear framework with iterative refinements to boost sample fidelity. Additionally, extending the framework to higher-dimensional or multimodal datasets and integrating it with latent diffusion models could further amplify its impact, fostering scalable and efficient generative modeling solutions for real-world applications.

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
