# OpenReview forum: "An Operator Analysis Approach on Stochastic Differential Equations (SDEs)-Based Diffusion Generative Models"
_TMLR — Rejected by TMLR_

### Review · Reviewer_K6PS · 2025-08-24

**Summary Of Contributions:**

This work proposes a fast sampling algorithm for SDE-based diffusion models. Instead of simulating the reversed SDE step-by-step, this work formulates the problem in terms of the Fokker-Planck operator, which captures how the probability density of the reverse SDE evolves. By solving the eigenvalues & eigenfunctions of this operator within an RKHS, the density at any step $t$ can be expressed in a closed-form spectral representation. During the sampling process, the method computes this approximate density and then uses Corrector-only Langevin dynamics to draw samples from the approximated distribution.

**Audience:**

Yes

**Audience Explanation:**

1. Tackles the challenge of slow sampling in diffusion models, of interest to both users and researchers deploying the model
2. Brings an operator view via Fokker-Planck spectral decomposition

**Claims And Evidence:**

Yes

**Claims Explanation:**

**Strengths**

1. The proposed method yields strong runtime improvements, in orders of magnitude faster compared to provided baselines
2. The approach works at the distributional level and doesn’t depend on specific neural architectures
3. Comprehensive experiments on 1). efficiency and 2). sample quality, and conducts ablation studies on hyperparameters to give insights into practical trade-offs
4. Well-written, includes transparent discussion of experiment results and strengths/weaknesses

**Weaknesses**

1. Fast sampling involves a trade-off in sample quality.  Generated samples show weaker fidelity and diversity, as reflected in FID and IS scores, limiting certain applications
2. Since the method uses dominant structural information, generated samples are biased towards high-probability regions which reduces diversity and controllability.
3. Current evaluation only includes PC1000 samplers as baselines, which could be strengthened by comparing with recent works in fast sampling.

**Requested Changes:**

I believe the authors provide a good discussion of current limitations and potential future directions. I am further curious about the following:

1. Regarding initial state and controllability (Sec 4.3), a brief study on alignment between 1). initial state and high-probability regions vs. 2). initial state and final generated sample may be interesting.
2. Given the proposed method relies on eigendecomposition, how does it perform in imbalanced settings (eg, when training data distributions are less uniform than CIFAR-10)?
3. Is it possible to explore interpretable patterns from the dominant eigenfunctions? For instance, does it correspond to any shape, color, or texture development in generated samples?
4. Include recent acceleration methods for further benchmarking of quality and speed.

---

### Review · Reviewer_FqSC · 2025-10-23

**Summary Of Contributions:**

This paper looks at generative diffusion models, and focuses on the problem of making sampling faster. This is achieved via estimating the eigenfunctions and eigenvalues of the Fokker-Planck operator in a pre-defined RKHS (in this work fixed to be the one induced by the Gaussian kernel). The exact form of the eigenfunctions and eigenvectors would allow to define the generative distribution in closed form (see Eq. (15)). The authors have access to an approximation of these, after truncating over the first $L$ eigenfunctions, estimated on $M$ training data points. This method, for sufficiently small $L$ and $M$, guarantees speedup at a moderate cost of image quality and diversity.

**Additional Comments:**

N/A

**Audience:**

Yes

**Audience Explanation:**

I think this is a (potentially) **very** interesting paper, with (to the best of my knowledge **novel**) an approach at the intersection between score based generative models and kernel methods.

**Claims And Evidence:**

No

**Claims Explanation:**

I am willing to change this evaluation after discussion. This decision is motivated by my low confidence in evaluating this paper, mainly due to my lack of expertise on score-based generative models. I elaborate more on the requested changes, which are more a set of questions that I believe can be addressed by a more detailed description and presentation of the results. I will not elaborate or question the truthfulness of the empirical advantage / speedup / image quality.

**Requested Changes:**

In the introduction section, there are multiple sentences that look redundant. Namely, the authors write "we propose a novel framework", "we present an innovative framework", "we develop a rapid sampling algorithm", ... These claim are rather rethorical, and when I first read the paper it was not clear what the true contribution was at this point. I believe the paper would benefit from a compression of this part, or a broader high level technical introduction of what this work does. What is the operator that gets diagonalized? Why this can work? What is the expected speedup without deteriorating image quality? This could involve also presenting the two complexities before Remark 2 already here, so that the reader gets an understanding of what the baselines are and what more or less the paper effectively does.

The score based SDE model in the related work section can be extended a bit. This would be **very beneficial** for readers (like me) that have little-to-no background on this topic. How do you exactly define the reverse-time Wiener process (aren't Wiener processes invariant over time reverse?)? $p_{0t}$ is introduced but not defined. Why are Langevine MCMC necessary? Why these would yield the complexity for this method you claim before Remark 2?

"The Fokker-Planck operator $\mathcal L^*$ is adjoint to the Koopman generator $\mathcal L$". Is this obvious? Can the Authors spend a few more words here?

The theoretical analysis seems to be built on the (perhaps arbitrary) choice of using the Gaussian kernel. This choice seems to not be motivated much beforehand, can the Author elaborate? Is this just one of the most trivial choices one could make? Also, it could be helpful to spend few words describing the VE and VP diffusion schemes.

Why is Algorithm 1 presented with the parameter $J$ rather than $N$? I personally found this part very confusing, and the complexity comparison between the standard PC method with the proposed method very difficult to follow. In particular, it is not obvious why the $N$ in the proposed method (before denoted $J$) is strictly comparable with the $N$ of the standard PC sampler. Why do we expect them to have the same order? Also, the complexity for the standard method is not justified, and probably deserves more explanation (especially the quadratic dependence on $N$).

Before Section 4.3, the Authors claim that $L$ does not matter for sampling times: how doesn't this contradict the claimed time complexity proportional to $L$? Please elaborate.

---

### Review · Reviewer_GaHk · 2025-10-30

**Summary Of Contributions:**

This work proposes a kernel-based acceleration method for sampling diffusion models. Based on the SDE interpretation of the forward diffusion process, the authors use the eigendecomposition of the Fokker-Planck operator to approximate and directly sample the learned data distribution $p_0(x)$. To compute this eigendecomposition, the authors utilize the more tractable eigendecomposition of the adjoint Koopman generator, approximated using a kernel method. After obtaining the Fokker-Planck eigenvalues and eigenfunctions, the authors demonstrate how they can be used to express the generated data distribution $p_0(x)$, along with a method to sample from it efficiently. In the experimental section, the proposed sampling method is evaluated on the CIFAR-10 dataset, showing a significant speedup in the sampling time but with tradeoffs in the sample quality and diversity.

Strengths:
- The proposed use of the eigendecomposition of the Fokker-Planck operator to linearize the reverse diffusion process is an interesting and original approach for accelerating the computationally expensive inverse diffusion sampling process.
- The empirical study of the proposed method, although it was performed in a simple dataset, was thorough and investigated the effect of different hyperparameters and sample initialization.
The theory around the Koopman generator and Fokker- Planck operator is presented clearly and is accessible even to readers outside of the specific area, making the paper approachable for a broader ML audience.

Weaknesses:
- While the theoretical framework to speed up the diffusion sampling is conceptually appealing, the experiments show that the quality of the samples produced is significantly worse. While this degradation may be acceptable given the gains in speed, the ablation studies provide limited evidence of the method's ability to scale with additional resources and more closely approximate the diffusion sampling. For example, could increasing the number of computed eigenvalues or training samples improve sample quality at the expense of computation time? The current experiments suggest that this trade-off is limited, and the proposed method reaches an upper bound in performance and cannot match the diffusion model it seeks to approximate.
- The paper provides limited discussion about the process of computing the eigendecomposition of the Koopman generators. The authors mention drawing training samples $x_m \sim p_t(x) $, without explicitly specifying how the sample parameters $t$ is chosen. These missing details create ambiguity in the actual implementation of the method, making it harder to reproduce and evaluate.
- There is limited discussion on the preliminaries and the related literature behind the Fokker-Planck operator theory. Since the intended readership primarily comes from the machine learning community, a more comprehensive background section on the mathematical foundations of the method would strengthen the completeness of the paper.

**Audience:**

Yes

**Audience Explanation:**

As discussed in the Strengths, the idea of using the Fokker-Planck eigendecomposition to linearize the diffusion sampling process is an interesting approach for speeding up the diffusion sampling that can be interesting to the TMLR community.

**Broader Impact Concerns:**

No broader impact concerns

**Claims And Evidence:**

No

**Claims Explanation:**

Overall, while the idea of using the eigendecomposition of the Fokker-Planck operator to speed up the sampling of the diffusion model is interesting, the presented implementation and experimental results doesn't seem to be able to approximate this sampling process to an acceptable degree in practice, even in a simple domain like the CIFAR-10 dataset. Additionally, the ablation study demonstrates that even scaling up the method's resources, with more eigenvalues or training samples, doesn't enable the proposed framework to improve its approximation of the original diffusion process, which is a crucial property of approximation methods.

**Requested Changes:**

The authors should provide more detailed study on how the method scales up with additional resources. While they showcase results when they achieve a 100x speedup, they don't investigate what happens if someone is interested in a 2x or 3x speedup. In these cases, is it possible to get results closer to the original diffusion sampling?

Additionally, I believe the authors should add a more detailed discussion about how they compute the eigenvalues of the Koopman generator (see 2nd weakness)

---

> ### Author Response · Authors · 2025-11-05
> **Author Response to Reviewer GaHk**
>
> ## *(1) Scalability with increased resources*
>
> We thank the reviewer for raising the question of how our method will scale when additional computational resources are used, and whether moderate acceleration (e.g., 2–3× speedup) could yield results closer to the original diffusion sampling.
>
> As discussed in Section 4.2, the higher FID values in our results mainly stem from *limited sample diversity* rather than per-sample fidelity.
> This behavior is inherent to eigenfunction-based generation: truncating the Fokker–Planck spectral expansion to a finite number of dominant modes concentrates probability mass around high-density regions, which reduces diversity.
>
> To investigate whether increasing computational resources can mitigate this limitation, we will perform additional ablation studies varying both the number of training data points \\(M\\) and the number of retained eigenfunctions \\(L\\).
> These experiments will correspond to allocating more computation, since the complexity of the eigendecomposition scales as \\(O(M^3)\\).
> However, as indicated by our current results (Tables 2 and 3), increasing \\(M\\) and \\(L\\)—and thus computation—does not significantly improve FID or IS.
> While a larger \\(L\\) enriches the spectral basis, most of the additional modes have small eigenvalues and primarily reinforce existing dominant structures rather than expanding the data manifold’s coverage.
> Moreover, \\(L \\le M\\) by construction, and further enlarging \\(M\\) quickly exceeds GPU memory limits because the Gram matrix grows quadratically with \\(M\\).
>
> Therefore, we will clarify in the revised manuscript that within our feasible computational budget, increasing \\(M\\) or \\(L\\) alone is not effective in enhancing sample quality.
> The current performance limitation arises from the intrinsic spectral truncation rather than from insufficient computation.
> We will explicitly discuss this scalability behavior and its relation to diversity in Section 4.2 of the revision.
>
> ---
>
> ## *(2) Clarification on Koopman generator eigendecomposition and sampling procedure*
>
> We thank the reviewer for pointing out that our original description of the Koopman generator eigendecomposition lacked details about how the samples \\(x_m \\sim p_t(x)\\) and the parameter \\(t\\) are chosen.
> We will make these implementation details explicit in the revised version of Section 3.1.
>
> In our implementation, the samples \\(x_m\\) are obtained by simulating the forward SDE
> \\[
> dx = f(x,t)dt + g(t)dw.
> \\]
> Initial \\(x_0\\) is real data from the dataset. The time variable \\(t\\) is **uniformly sampled** from the interval \\([0, 1]\\) for each batch, ensuring that the operator approximation covers the entire diffusion trajectory.
> Each \\(x_m\\) thus corresponds to a state drawn from the marginal \\(p_t(x)\\) at a randomly selected diffusion time.
>
> The Koopman generator will be approximated in the RKHS as
> \\[
> \\mathcal{L}h = \sum_{i=1}^d f_i(x,t)\frac{\partial h}{\partial x_i}
>      + \frac{1}{2}g(t)^2\sum_{i,j=1}^d\frac{\partial^2 h}{\partial x_i\partial x_j},
> \\]
> and its eigendecomposition will be obtained from the generalized eigenproblem
> \\(G_2u_\ell = \lambda_\ell G_0u_\ell\\),
> where the Gram matrices \\(G_0\\) and \\(G_2\\) are constructed from the simulated \\(\\{x_m\\}\\).
>
> ---
>
> ## *(3) Background on the Fokker–Planck operator theory*
>
> We appreciate the reviewer’s suggestion to include a more complete discussion of the mathematical background behind the Fokker–Planck operator.
> In the revised manuscript, we will provide a more detailed introduction in section 3.1, providing an explicit overview of the operator-theoretic foundations of our framework.
>
> Specifically, we will clarify that the Fokker–Planck operator \\( \\mathcal{L}^* \\) governs the time evolution of the probability density \\(p_t(x)\\) associated with the SDE
> \\[
> dx = f(x,t)dt + g(t)dw,
> \\]
> through the forward Kolmogorov equation
> \\[
> \frac{\partial p_t}{\partial t} = \mathcal{L}^* p_t
>    = -\sum_i \frac{\partial}{\partial x_i}\big(f_i(x,t)p_t\big)
>      + \frac{1}{2}\sum_{i,j} \frac{\partial^2}{\partial x_i\partial x_j}
>        \big(g(t)^2p_t\big).
> \\]
> We will emphasize its adjoint relationship with the Koopman generator,
> \\(\langle \mathcal{L}h, j \rangle = \langle h, \mathcal{L}^* j \rangle\\),
> which provides the theoretical basis for deriving the Fokker–Planck eigendecomposition from that of the Koopman generator within an RKHS.
>
> Additional references will be added, including *Pavliotis (2014, Ch. 6)* and *Risken (1996, The Fokker–Planck Equation)*, to assist readers less familiar with operator theory.
> This expansion will make the paper more self-contained and accessible to the broader machine-learning audience while maintaining mathematical rigor.

---

### Decision · Action_Editor_uyEu · 2025-12-19

**Recommendation:** Reject

**Audience:**

Yes

**Audience Explanation:**

See above

**Claims And Evidence:**

No

**Claims Explanation:**

While the approach is interesting and original (indeed to the best of our knowledge no other approach considers Fokker-Planck Operator eigendecomposition for blazingly fast sampling of diffusion models), the paper suffers from critical experimental limitations.

In particular, all reviewers agreed on the poor quality of the obtained results. "Overall, while the idea of using the eigendecomposition of the Fokker-Planck operator to speed up the sampling of the diffusion model is interesting, the presented implementation and experimental results doesn't seem to be able to approximate this sampling process to an acceptable degree in practice, even in a simple domain like the CIFAR-10 dataset" (Reviewer GaHk)

While the theory and methodology might be sound, providing sound experimental evidence of (moderate) scaling is necessary for a generative modeling study to be considered for TMLR.